# Evidence-Based Practices to Prevent Ventilator-Associated Pneumonia in an Intensive Care Unit in Bangladesh

**DOI:** 10.3390/healthcare13212782

**Published:** 2025-11-02

**Authors:** Nahida Akhter, Xintong Zhou, Sameh Elhabashy, K. A. T. M. Ehsanul Huq, Md Moshiur Rahman, Michiko Moriyama

**Affiliations:** 1Graduate School of Biomedical and Health Sciences, Hiroshima University, Hiroshima 734-8553, Japan; 2Faculty of Nursing, Cairo University, Cairo 12613, Egypt

**Keywords:** ventilator-associated pneumonia, intensive care units, incidence, evidence-based practice, nursing care

## Abstract

Background: Ventilator-associated pneumonia (VAP) is a major cause of morbidity and mortality in intensive care units (ICUs), particularly in low- and middle-income countries (LMICs). Evidence-based practice (EBP) bundles have shown effectiveness in reducing VAP; however, the implementation in Bangladesh remains limited. This study aimed to evaluate the effectiveness of EBP to reduce the incidence rate of VAP among adult ICU patients in Bangladesh. Methods: A quasi-experimental study with a historical control group was conducted among 347 eligible ICU patients from October 2024 to April 2025. The intervention included nurse training on VAP bundle practices with advanced equipment support. Data on VAP incidence as a primary endpoint and VAP-related patients’ outcomes were analyzed. Results: The clinically suspected VAP incidence was 30.1 and 51.1 per 1000 ventilator-days, and the prevalence decreased significantly in the intervention group compared to the control group (26.9% vs. 46.1%; *p* < 0.001), respectively. Logistic regression indicated VAP bundle implementation was associated with reduced VAP (Exp(B) = 0.417, 95% CI: 0.262–0.666), while ventilation ≥96 h was a significant risk factor (Exp(B) = 2.6, 95% CI: 1.385–4.881). Early-onset VAP was reduced (25.0% vs. 10.2%), though late-onset predominated in the intervention group (75.0% vs. 89.8%). Conclusion: Implementation of an EBP-based VAP bundle by trained nurses significantly reduced VAP incidence. However, increased overall ICU mortality highlights the need for broader critical care improvements, including advanced comorbidity management and comprehensive ICU services. This study underscores the feasibility and effectiveness of VAP bundle implementation in the ICU of an LMIC.

## 1. Introduction

Ventilator-associated pneumonia (VAP) is one of the most frequent and severe hospital-acquired infections among patients in intensive care units (ICUs). The pathogenesis of VAP is often linked to the aspiration of colonized oropharyngeal secretions that bypass the endotracheal tube cuff. VAP is defined as developing after 48 h of intubation [1]. VAP is strongly associated with prolonged mechanical ventilation and extended ICU stays and causes substantial mortality [2]. Globally, the incidence of VAP ranges from 10% to 28%, with attributable mortality between 9% and 27% [3]. The burden is disproportionately higher in low- and middle-income countries (LMICs), where limited resources, overcrowded ICUs, and low nurse-to-patient ratios contribute to elevated infection rates [4].

The incidence of VAP is expressed as the incidence density per 1000 ventilator-days, which adjusts for differences in exposure time [5]. It provides a standardized metric that facilitates comparison across ICUs with varying caseloads and ventilation durations [6].

Evidence-based practice (EBP) bundles are widely recommended to prevent VAP. These bundles combine several simple interventions that work best when applied together and consistently [7]. Studies have shown that applying these bundles reduces VAP rates, shortens ventilation and ICU stay, and lowers mortality [8,9].

The diagnosis of VAP is challenging, as multiple guidelines exist with varying criteria for clinical features (fever ≥ 38 °C, purulent secretion, or leukocyte), chest radiological and microbiological (tracheal culture) confirmation by physicians following international guidelines [8,10,11,12,13]. While practical and aligned with prior studies in resource-limited ICUs, it may overestimate VAP incidence. Conversely, microbiological and radiological confirmation was not consistently available due to limited diagnostic facilities, as patients often bear the costs themselves. This may lead to underdiagnosis of subclinical or atypical cases, thereby affecting diagnostic accuracy. Consequently, the reported incidence reflects clinically suspected VAP rather than strictly confirmed cases [14].

Nurses with EBP training and preventive measures, such as using protective devices and hand hygiene practice, can reduce the VAP incidence in the ICU settings [15]. A systematic review and meta-analysis conducted by one study found that an open-suction catheter was associated with a 57% higher incidence of VAP compared to a closed-suction catheter [16]. Evidence from tertiary care ICUs in Bangladesh also indicates markedly lower survival rates among VAP-positive patients (31.4%) compared with VAP-negative patients (68.6%) [8]. Furthermore, microbiological surveillance reveals that *Acinetobacter baumannii*, *Pseudomonas aeruginosa*, and *Klebsiella pneumoniae* are the predominant pathogens, often exhibiting alarming multidrug resistance patterns, including carbapenem resistance [17]. These resistance trends further constrain therapeutic options and pose significant challenges for effective management.

In Bangladesh, VAP remains an underexplored yet pressing healthcare challenge. National-level statistics for ICU quality indicators are lacking, and available evidence is largely limited [7]. Several interventional studies have evaluated the effect of implementing an EBP ventilator bundle on adult ICU patients [18]. Most existing research remains descriptive, focusing on microbial patterns or baseline incidence rates, without exploring how structured, evidence-based preventive measures could improve outcomes [19]. This represents a significant knowledge gap, as understanding the local epidemiology is crucial for developing contextually appropriate and effective interventions. Given the high burden of VAP, the predominance of multidrug-resistant organisms, and the limited interventional research in adult ICUs in Bangladesh, there is a critical need to investigate the effectiveness of structured evidence-based interventions [20]. Therefore, this study aimed to evaluate the effectiveness of EBP by implementing VAP bundles to reduce the incidence rate of VAP among adult ICU patients in Bangladesh.

## 2. Materials and Methods

### 2.1. Trial Design

A quasi-experimental, non-randomized, two-arm study design with a non-equivalent historical control group was conducted, comparing two sequential time periods. As an intervention component, we provided the nurses with evidence-based training between the control and intervention periods. This design was selected due to the difficulty of dividing patients into distinct groups, as Dhaka Medical College Hospital (DMCH) comprises nine ICUs with varying levels of care and patient types, and nurses are frequently rotated among them.

The study was conducted from October to November 2024 for the control period and from February to April 2025 for the intervention period (61 days for each period). This study followed the Transparent Reporting of Evaluations with Nonrandomized Designs (TREND) [21] (Appendix A) and was registered at ClinicalTrials.gov (Identifier ClinicalTrials.gov-NCT06624540, registration date 12 September 2024).

A conceptual Framework of our study is illustrated in Figure 1.

In this study, the high incidence of VAP in the ICU was the primary problem trigger, combined with nurses’ limited knowledge, poor practices, and lack of resources, which are hypothesized as contributing factors. Considering VAP as a leading cause of morbidity and mortality, a multidisciplinary team was formed to address these issues. Evidence from international guidelines, including the Institute for Healthcare Improvement (IHI), the Japanese Society of Intensive Care Medicine (JSICM), and the Centers for Disease Control and Prevention (CDC), was appraised to develop a feasible VAP prevention bundle [8,10,12]. The intervention included training nurses, providing essential equipment, and implementing a bundle to control the contributing factors. Outcomes were monitored with a focus on both primary and secondary endpoints. If a significant improvement could be achieved, the intervention can be adopted and integrated into routine ICU practice. This framework highlights a structured approach for applying EBP in resource-limited ICU settings to improve patient outcomes.

### 2.2. Study Setting and Participant Sample

The study was conducted in DMCH, a leading tertiary-level government hospital located in the capital city of Bangladesh. DMCH is a 2600-bed facility and houses nine distinct ICUs, accommodating a diverse range of patients across its 145 ICU beds, staffed by 202 nurses. We selected a general ICU where different types of critical patients such as medical and surgical were admitted. The ICU consists of 32 beds, 56 physicians and 54 nurses worked during the time of the study period; the nurse and patient ratio was 1: 3–4.

As an exploratory design, we considered the estimated sample size based on the monthly patient admissions within the study time period. In February 2024, a total of 115 patients were admitted to this ICU; therefore, we estimated that about 230 patients would be enrolled for the 2-month study period (estimated 450 patients for both groups).

We enrolled all the patients who were admitted during the study period after checking the eligibility criteria. Patients intubated after being admitted to the ICU, aged 18 years and older, irrespective of sex, and legal guardians of the patients who provided written informed consent to participate in this study were included. We excluded those who stayed less than 2 days in ICU, who died or were intubated and extubated within 2 days after admission and readmission to ICU.

To diagnose VAP, various studies employ different approaches, depending on the research setting and available resources. In our study, we applied the definition of clinically suspected VAP, as we considered it was more appropriate for this ICU setting in Bangladesh, where diagnostic facilities and medical records are limited [13]. We defined clinically suspected VAP with a new or progressive infiltration on chest radiography, along with sign of infection or clinical deterioration, such as fever (temperature >38 °C), or leukopenia (WBC counts: <4000 cells/mm^3^)/leukocytosis (WBC count: >12,000 cells/mm^3^, and a positive microbiological culture (tracheal or blood), after 48 h of intubation [22].

### 2.3. Outcomes

Primary endpoint

The primary endpoint determined the incidence rate of VAP. The incidence rate of VAP was calculated:$$\frac{The\;number\;of\;VAPs\;for\;a\;location}{The\;number\;of\;ventilator\;days\;for\;that\;location}\times 1000$$

A diagnosis of clinically suspected VAP was confirmed by ICU physicians and followed by Kalil et al. (2016) [13]. All mechanically ventilated patients were monitored daily for clinical indicators of VAP throughout the study period [13].

Secondary endpoints

The secondary endpoints were: (1) the interval from ICU admission to VAP occurrence, (2) the interval from intubation to VAP occurrence, (3) the micro-organism link to VAP, (4) the mortality rate of ventilated patients in ICU, (5) early onset (<96 h) and late onset (≥96 h) of VAP, and (6) the average length of stay of ICU. We also collected socio-demographic data including age, sex, and hospital admission information such as days in ICU, days before ICU after admission in the hospital, and ventilation days.

Nurses’ skill on the VAP bundle was concurrently assessed using a researcher-developed skill checklist, the validity of which was confirmed prior to the study. Content validity was checked following Lynn’s method [23]. Each item was assessed by three independent trained critical care nurses for appropriateness, clarity, and relevance of the checklist. The necessary changes were made in response to their input. The final version of the checklist had a maximum score of 140, including 64 VAP care-related items. Operationally, the performance levels were divided into three categories: low ≤69/140 score (<50%), moderate 70–111/140 score (50–79%), and high ≥112/140 score (≥80%). Content validity was shown as ‘Excellent’ by the scale-level content validity index based on universal agreement (S-CVI/UA) of 0.97 and the item-level content validity index (I-CVI). Cronbach’s alpha, which measures internal consistency dependability, produced a coefficient of 0.84, indicating acceptable reliability.

During the pre-intervention period, baseline assessments of nurses’ skills were conducted using this checklist. Following a two-month training period on the VAP bundle, nurses’ skills were reassessed during the intervention period by research assistants (RAs).

The checklist comprised ten core components of the VAP prevention bundle, with each task rated as “not performed” (0), “unsatisfactorily performed” (1), or “satisfactorily performed” (2). RAs evaluated nurses’ performance in real-time clinical settings, carefully observing task completion and marking the checklist accordingly to ensure accurate and objective assessment.

### 2.4. Intervention

We implemented the VAP bundle as an intervention by trained ICU nurses with the introduction of advanced equipment.

Implementation of VAP bundle components

The VAP prevention bundle implementation training was designed by the international guideline based on the IHI, 2024; JSICM-VAP Bundle, 2023; and CDC, NHSN [8,15,16]. The VAP bundle consisted of ten evidence-based components: maintain five moments for hand hygiene, head-of-bed elevation at 30°–45°, oral care with chlorhexidine three times/day, avoidance of oversedation, proper breathing circuit management (unnecessary circuit change), maintenance of endotracheal tube cuff pressure using a manometer, use of closed tracheal suctioning system and subglottic suctioning, daily assessment for weaning, early mobilization, prophylaxis for peptic ulcer disease and deep vein thrombosis (DVT) (using DVT stocking and regular examine). Collectively, these measures represent essential elements of care for patients receiving mechanical ventilation.

Nurses’ training on VAP bundle

The training was designed to improve nurses’ cognitive and psychomotor performance related to VAP prevention in the ICU. At the beginning of the program, the researcher introduced the concept of EBP and raised nurses’ awareness. The educational content was organized into lectures and skills-based practice that addressed different components of the VAP prevention bundle, developed in accordance with the CDC guidelines. The training was provided for two months with four sessions followed by daily monitoring.

Equipment used in intervention period for preventing VAP

To prevent VAP, advanced equipment was introduced and substituted for routinely used devices. The equipment used reflected current EBP globally, such as closed suction catheters to prevent cross-infection, endotracheal tubes, and suction devices. As an EBP, we also used disposable gowns and oral care brushes, which were not used in this hospital. We used global standard equipment to ensure patient safety and prevent VAP.

### 2.5. Control Group

Participants received the usual care routinely provided in the ICU for mechanically ventilated patients. DMCH has a scarcity of manpower, logistics and supplies due to the overburden of patients and is considered one of the low-resource institutions in Bangladesh. In the existing care system, there were no standard protocols followed, and no standard equipment was used for the prevention of VAP. For example, an open suction system was employed instead of a closed suction, and a single suction catheter was frequently used for longer than a week. Cuff pressure was not measured by any system. Oral care was provided only once per day. Insufficient awareness and a lack of personnel lead to irregular patient repositioning. Additionally, during the time on mechanical breathing, defined weaning procedures and sedation vacation guidelines were not regularly observed. We observed patients and collected data for documentation.

### 2.6. Data Collection Procedure

RAs (*n* = 10), who were registered nurses in Bangladesh were selected by the principal investigator (PI) and Director of Nursing. They had been working in different departments, including administrative, general, and specialized units such as the ICUs. They also had clinical experience in infection control and critical care units. The PI and the research team (3 persons including an ICU physician) provided one week of training to RAs on the use of EBP and the VAP bundle. They were responsible for delivering evidence-based education, facilitating bedside teaching, and conducting daily patient monitoring. Their roles were to educate, teach, and monitor the participatory nurses. These RAs assisted participatory nurses for patient care, taking feedback on the challenges they faced and resolving problems that arose during patient care. The RAs provided support to the nurses, ensuring implementation of EBP in the ICU. They also played a critical role in patient observation and ensuring sustained engagement with the intervention throughout the study period.

After patients were admitted to the ICU, RAs obtained consent and collected clinical and laboratory data from the patients’ records. They collected data on the interval between ICU admission and VAP occurrence, the interval from intubation to VAP occurrence, the mortality rate of mechanically ventilated patients in ICU, early-onset (<96 h) and late-onset (≥96 h) of VAP, and the average length of stay of patients in ICU. Then, data were input into a password-protected computer in a standardized Excel sheet on a daily basis. Patient outcomes and infection rates were recorded for both groups. Patients’ data, including demographic characteristics, duration of ICU stay, length of stay in the ICU, outcomes during ICU admission, duration of mechanical ventilation, and relevant clinical information (medical history, laboratory test results, and diagnoses) were collected.

### 2.7. Quality Management

PI and the research team periodically monitor RAs’ performance, including consent-taking and data collection procedures and change the procedures where necessary. They held regular meetings with the RAs to assess the data quality, participant recruitment, accrual and retention, participant risk versus benefit, performance and resolve any confusion and concerns. RAs checked the quality control of the participatory nurses, verifying the data collected from participants on the same day. The RAs consulted with the PI and the research team about any queries, and they examined the information and answered the questions. All patients received equitable monitoring and immediate intervention for any complications, ensuring no compromise to their safety or quality of care during the study period.

### 2.8. Statistical Analysis

All statistical analyses were performed using SPSS (version 29.0, IBM Corp., Armonk, NY, USA). Continuous variables were expressed as mean and standard deviation (SD) and compared between groups using independent-sample *t*-tests. Categorical variables were presented as frequencies and percentages and compared using Chi-square tests or Fisher’s exact tests where appropriate. Clinically suspected VAP incidence was calculated as the number of VAP cases per 1000 ventilator days. Early- and late-onset clinically suspected VAP were classified according to onset time relative to initiation of mechanical ventilation. Binary logistic regression analysis was conducted to identify independent predictors of clinically suspected VAP. Candidate covariates included sociodemographic, clinical, and intervention-related variables that were considered clinically relevant. Interaction terms were also examined where appropriate. Results were expressed as regression coefficients (B), standard errors (SE), *p*-values, odds ratios [Exp(B)], and 95% confidence intervals (CI). Model fit was evaluated using the Chi-square test of overall model significance. A two-tailed *p*-value < 0.05 was considered statistically significant.

## 3. Results

A total of 227 and 217 patients were admitted to the ICU, and 165 and 182 patients were enrolled and analyzed in the control and intervention groups, respectively. Additional information regarding the reasons for dropout is presented in Figure 2.

There were no statistically significant differences observed between the two groups with respect to age, sex, duration of ventilation, days before ICU admission, or length of stay in ICU (Table 1).

Table 2 shows that the number (rate) of clinically suspected VAP and per 1000 ventilator days decreased significantly (76 (46.1%) to 49 (26.9%), *p* < 0.001; 51.1 to 30.1, respectively). Although there was no statistically significant difference in the proportion of patients ventilated for less than or ≥96 h between the control and intervention groups, the incidence of early-onset clinically suspected VAP decreased significantly from 25.0% to 10.2% (*p* = 0.040). The interval from ventilation to clinically suspected VAP occurrence did not differ significantly (*p* = 0.636) between the groups. Although the mortality rate among patients with clinically suspected VAP did not change significantly (*p* = 0.082) between the two groups, the overall ICU mortality rate among ventilated patients significantly increased from 50.9% to 62.6% after the intervention (*p* = 0.028).

Table 3 shows the causes of mortality by major organ systems (Fisher’s exact test = 12.716, *p* = 0.157), and the differences in deaths were more for neurological, pulmonary and renal causes in the intervention group than the control group.

The binary logistic regression analysis presented in Table 4 indicates that patients in the intervention group had significantly lower odds of clinically suspected VAP (B = −0.874, *p* < 0.001, Exp(B) = 0.417; CI 0.262–0.666), indicating that these patients had 41.7% lower odds of clinically suspected VAP compared with the control group.

The intervention group showed a higher rate of positive culture (tracheal and blood) (77.9% vs. 40.7%) among the clinically suspected VAP cases, and *Acinetobacter* was found as the most detected isolate, followed by *Pseudomonas* and *Klebsiella*. Details of the distribution of micro-organisms in ventilated patients and those with clinically suspected VAP were provided in Appendix A.

The number of ICU days was also a significant predictor (B = 0.025, *p* = 0.022, Exp(B) = 1.025; CI 1.004–1.047), with each additional day in the ICU increasing the odds of clinically suspected VAP by 2.5% among ventilated patients. Although general ventilation days alone were not a significant predictor ((B = −0.015, *p* = 0.466, Exp(B) = 0.985; CI 0.945–1.026), ventilation for 96 h or more was significant and positive (B = 0.956, *p* = 0.003, Exp(B) = 2.6; CI 1.385–4.881), suggesting that after 96 h of ventilation, patients faced a 2.6 times risk of clinically suspected VAP than before 96 h. Overall, the model was statistically significant (Chi-square = 36.530, *p* < 0.001).

ICU nurses’ skill levels in implementing the VAP bundle improved significantly after the training (*p* < 0.001). Detailed compliance with each of the ten VAP bundle items pre- (control period) and post- (intervention period) training is provided in Appendix A. 

## 4. Discussion

To the best of our knowledge, this is the first study to examine the incidence of VAP in adult patients and to apply an EBP bundle to prevent VAP as the primary comorbidity in the ICU in Bangladesh. The study introduced a VAP prevention bundle with advanced equipment and was implemented by EBP-trained nurses. Although the sample sizes were not equal between the control and intervention groups, and randomization was not applied, no statistically significant differences were observed in patient characteristics between the two groups. This may be attributed to the “dynamic equilibrium” often observed among ICU patients [24]. Overall, the non-equivalent design did not introduce confounding variables.

Following the intervention, indicators associated with the incidence of clinically suspected VAP improved, and binary logistic regression identified “intervention” as a major protective factor. These findings demonstrate the success of this study in reducing VAP incidence among ICU patients through the implementation of EBP using the VAP bundle. The study nurses received EBP training and monitored during bedside patient care, which was the effective implementation of EBP. These actions were intended to improve the ICU critical care services and enhance adherence to EBP. The discussion below highlights the key findings, EBP implications, and areas for future improvement.

The incidence of VAP among study participants decreased notably in the intervention group compared with the control group (30.1 vs. 51.1 per 1000 ventilator days). These findings highlight that even in low-resource settings, the introduction of EBP, advanced equipment, and active nursing supervision can substantially reduce VAP. Before the intervention, the clinically suspected VAP rate was quite high compared with other LMICs. For example, a systematic review and meta-analysis by Bonnell et al. (2018) reported a pooled VAP incidence of 18.5 per 1000 ventilator days in LMICs in Asia, nearly double that of high-income countries (9 per 1000 ventilator-days), largely due to limitations in infection prevention infrastructure, staffing, and prolonged ventilation [25]. Similarly, extremely high rates have also been documented, such as 43.7 per 1000 ventilator-days in Mongolia.

Nevertheless, under the challenging conditions, the intervention successfully reduced VAP incidence, both in terms of overall prevalence and incidence per 1000 ventilator days. This might support the effectiveness of VAP bundles implemented by trained nurses in controlling VAP incidence and is consistent with previous studies [26]. Binary logistic regression further confirmed that the implementation of the VAP bundle significantly contributed to the reduced onset of VAP. On the contrary, ventilation ≥96 h was identified as a major risk factor, consistent with earlier reports [27].

Even though clinically suspected VAP was statistically reduced, the total mortality of ventilated patients was increased in the intervention group. For that, we tried to explore the possible cause of deaths with clinical manifestations and laboratory diagnosis of the patients. We did not find any significant difference in tracheal culture and GCS scoring between the intervention group and control group (Figure 2). In the control group, patients performed a greater number of laboratory tests compared to the intervention group, it means that more patients in the intervention group did not perform laboratory tests, which might be an unknown cause of more deaths in the intervention group, which is very difficult to explain. We compared patients’ admission diagnoses and did not find any significant difference in disease criteria between the two groups. However, neurological, pulmonary and renal causes of death were found more in the intervention group compared to the control group. Moreover, a significantly higher proportion of late-onset VAP was associated with prolonged mechanical ventilation in the intervention group, along with limitations in overall patient management and challenges in reducing ventilation duration. The patients were enrolled at two different time points; therefore, seasonal infection trends could be a contributing factor for the higher mortality rate in the intervention patients. Supplementary microbiological data revealed the high prevalence of multidrug-resistant pathogens in the intervention patients, particularly *Klebsiella pneumoniae*, which are known to be associated with poor clinical outcomes and increased case fatality rates in critically ill patients. The limited availability of effective antimicrobial agents and delays in appropriate therapy likely contributed to the higher mortality observed during the intervention period. Therefore, to prevent complications, it is important to monitor and manage patients closely.

Although infection decreased significantly in the intervention group compared to the control group, other important aspects including advanced comorbidity management, organ support, and the availability of ICU services, might have remained the same. The complex nature of ICU treatment in LMICs is highlighted by this observation, as single-domain interventions may enhance certain outcomes but not overall survival. To investigate these important elements, more research is recommended.

While there was no ratio difference between patients ventilated for less than 96 h and those ventilated longer, the intervention group exhibited not only a reduction in overall VAP incidence but also a marked decrease in early-onset VAP (from 25.0% to 10.2%). However, no significant differences were observed in the interval from ventilation to clinically suspected VAP onset or in mortality among VAP patients between the control and the intervention groups. Conversely, in the intervention group, late-onset VAP increased nearly ninefold (89.8%) compared with early-onset cases. Though we could reduce VAP incidence with intervention, the patients had significantly higher rates of late-onset VAP, which may have contributed to the higher mortality in the intervention group. As it is more fatal; therefore, hospital infection control measures, including the management of antimicrobial resistance, need to be strengthened [28]. Moreover, the total mortality rate of mechanically ventilated patients significantly increased in the intervention. These results indicate that improvement is achievable not only through EBP alone, but also with comprehensive strategies [29].

This study initially hypothesized that nurses’ knowledge and skill levels contribute to VAP incidence, and the findings support this assumption. Although the specific mechanisms through which nursing competence influences patient outcomes remain unclear, a decline in VAP incidence was observed following skill enhancement. These findings provide a basis for future investigations into the role of nursing proficiency in patient outcomes.

Prior to the intervention, the incidence of clinically suspected VAP in this ICU was higher than that reported in LMICs and the global average [30], highlighting the limitations of healthcare resources in the study setting. This hospital is the central tertiary hospital that treats referral patients throughout the country. Therefore, this study findings can be applicable to other hospitals in Bangladesh, even in other LMICs, to strengthen the health system by implementing EBP through nursing training, availability of equipment and logistics.

Although the ICU is one of the most advanced care-providing units, the absence of a comprehensive health insurance system requires patients to bear ICU costs out of pocket, which typically restricts diagnostic testing to a single chest radiography and microbiological examinations [31,32].

Strengths

This study is the first to evaluate the incidence of VAP and the impact of EBP in the ICU in Bangladesh, which provides epidemiological data as a country’s reference for VAP. A quasi-experimental design of this study with a historical control group is one of its main strengths since it enabled an evaluation of the intervention in a high-burden ICU. Furthermore, multimodal training improved nurses’ skills and level of involvement.

Limitations

This study has certain limitations. As we did not calculate the sample size, it might cause underpowering of the study, thus a lack of generalizability. We used clinically suspected VAP instead of a confirmed diagnosis of VAP. The diagnostic criteria of VAP, the facility of diagnostic test, resource-limited setup, and clinician expertise are factors affecting the diagnosis of VAP. Moreover, more ventilated patients died in the intervention group compared to the control group where the cause of death was not possible to explore due to the scarcity of laboratory tests. Therefore, the study findings cannot be extrapolated to other hospitals in Bangladesh or other LMICs. Moreover, the timing of the controls and intervention differed, suggesting that some environmental factors such as seasonal variation may have contributed to the observed differences. The causal relationship between nurses’ training effect on patients’ clinical outcomes could not be established as the nurses conducted roster duties. All the study nurses were assigned to all the ICU patients (both intervention and control groups). The manual data collection method raises the possibility of recording errors. The lack of an electronic data system and limitations in diagnostic accuracy may have affected the reliability of the data.

## 5. Conclusions

We observed a substantial reduction in the incidence and prevalence of clinically suspected VAP following the implementation of a VAP prevention bundle by trained nurses using advanced equipment. Although the occurrence of late-onset VAP increased after intervention, no significant difference in mortality among patients with clinically suspected VAP was found between the intervention and control groups. However, overall ICU mortality was significantly higher after the intervention, which may be attributable to prolonged intubation and the increased late-onset VAP.

In this resource-limited setting, EBP alone was insufficient to reduce morbidity and mortality without comprehensive critical care management. Moreover, ventilator management and overall patient care remain significantly inadequate. Therefore, it is essential to train and educate intensive care specialists, nurses, and support staff, strengthening their competencies to improve clinical outcomes through sustainable training programs and the availability of essential laboratory facilities to ensure systematic and evidence-based critical care. To mitigate hospital-acquired infections, the government and policymakers must engage all relevant stakeholders to improve hospital infrastructure, logistical support, patient awareness, and the capacity building of healthcare personnel. A multifaceted approach incorporating advanced clinical management, EBP, and systemic reforms is essential.

Due to constrain on clinical and diagnostic facilities, we relied on clinically diagnosed VAP. Future intervention studies should employ randomized designs with larger sample sizes, extended follow-up periods, and robust clinical and diagnostic capabilities to more accurately identify the determinants of VAP.

## Figures and Tables

**Figure 1 healthcare-13-02782-f001:**
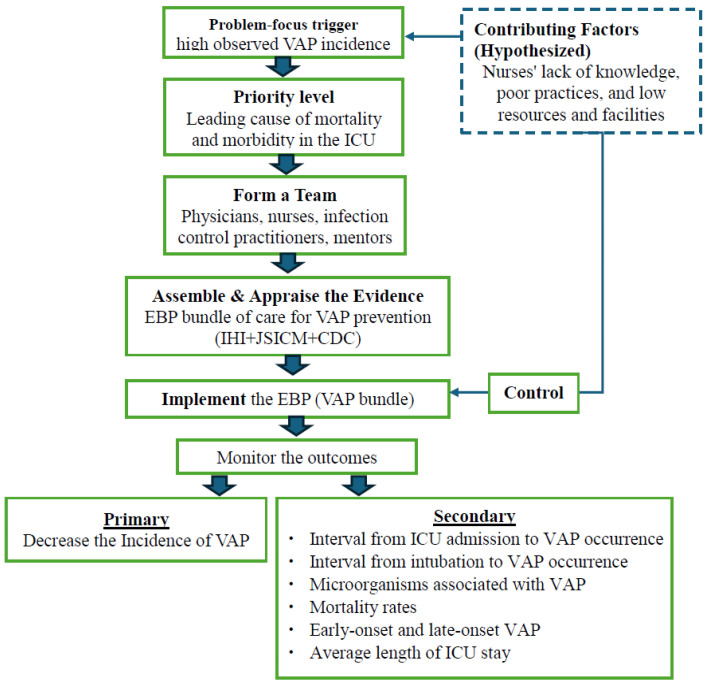
Conceptual framework.

**Figure 2 healthcare-13-02782-f002:**
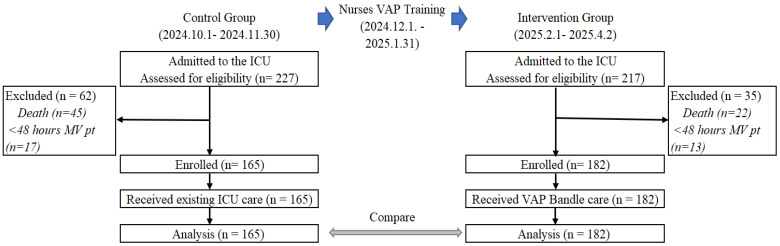
TREND study flow diagram. VAP-Ventilator-associated pneumonia, MV-Mechanical ventilator, ICU-Intensive care unit.

**Table 1 healthcare-13-02782-t001:** Sociodemographic characteristics and ICU stay of the ventilated patients (*n* = 347).

Sociodemographic	Control Group F (%), Mean (SD)	Intervention Group F (%), Mean (SD)	Chi-Square *t*-Test	*p*-Value
Ventilated patients	165	182		
Age (years)	45.6 (17.9)	45.0 (18.6)	0.349	0.727
Sex				
Male	119 (72.1%)	121 (66.5%)	1.290	0.256
Female	46 (27.9%)	61 (33.5%)		
Ventilation days	9 (7.4)	9 (7.0)	0.037	0.970
Days before ICU	4.4 (8.2)	4.5 (15.2)	−0.123	0.902
Days in ICU	12.7 (14.1)	10.8 (12.1)	1.319	0.188

F—frequency, SD—Standard deviation, ICU—Intensive care unit.

**Table 2 healthcare-13-02782-t002:** Ventilation and clinically suspected VAP status by study groups in ICU.

	Control Group *n*, F (%), Mean (SD)	Intervention Group *n*, F (%), Mean (SD)	Chi-Square *t*-Test	*p*-Value
Number of ventilation patients	165	182	—	—
Ventilation < 96 h	55 (33.3%)	56 (30.8%)	0.262	0.609
Ventilation ≥ 96 h	110 (66.7%)	126 (69.2%)		
Total ventilation days	1488	1630	—	—
Number of clinically suspected VAP	76 (46.1%)	49 (26.9%)	13.753	<0.001
Clinically suspected VAP episode per 1000 ventilator days	51.1	30.1	—	—
Early onset	19 (25.0%)	5 (10.2%)	4.204	0.040
Late onset	57 (75.0%)	44 (89.8%)		
Interval from ventilation to clinically suspected VAP occurrence	5.1 (7.9)	4.6 (3.4)	0.475	0.636
Tracheal culture (positive in total sample)	101/192 (52.6%)	93/163 (57.1%)	0.152	0.696
Blood culture (positive)	42/191 (22%)	22/156 (14.1%)	8.045	0.005
Mortality of clinically suspected VAP	36 (47.4%)	31 (63.3%)	3.027	0.082
Total Mortality of ventilated patients	84 (50.9%)	114 (62.6%)	4.858	0.028
GCS (death of ventilated patients)				
Score 3–8 (severe)	57/198 (28.8%)	76/198 (38.4%)	0.286	0.593
Score 9–15 (non-severe)	25/198 (12.6%)	28/198 (14.1%)		

VAP—Ventilator-associated pneumonia, ICU—Intensive care unit, F—Frequency, SD—Standard deviation, GCS—Glasgow Coma Scale.

**Table 3 healthcare-13-02782-t003:** Reasons of death in ICU main organ system involved.

Organic System	Total Deaths	Control Group (84)	Intervention Group (114)	Fisher’s Exact Test (*p*-Value)
Gastrointestinal	37	18	19	12.716 (*p* = 0.157)
Neurological	44	19	25	
Pulmonary	35	11	24	
Cardiovascular	4	2	2	
Renal	18	3	15	
Other causes				
Cesarean Section	4	1	3	
RTA	37	22	15	
Poisoning	7	3	4	
Carcinoma	5	2	3	

ICU—Intensive care units, RTA—Road traffic accident.

**Table 4 healthcare-13-02782-t004:** Binary logistic regression of the relations between clinically suspected VAP and selected variables.

Selected Variables	B	S.E.	*p*	Exp(B)	95% CI for Exp(B)	
					Lower	Upper
Group (post-intervention)	−0.874	0.238	<0.001	0.417	0.262	0.666
Days in ICU	0.025	0.011	0.022	1.025	1.004	1.047
Ventilation day	−0.015	0.021	0.466	0.985	0.945	1.026
Ventilation ≥ 96 h	0.956	0.321	0.003	2.6	1.385	4.881
Constant	−0.98	0.261	<0.001	0.375	—	—

Model fitness: Chi-square = 36.530, *p* < 0.05. VAP-ventilator associate pneumonia, CI—confidence interval, B—regression coefficient, S.E—Standard Error

## Data Availability

The dataset is available from the corresponding author upon request due to ethical considerations and the need to protect participant confidentiality. The data contains sensitive patient information, and according to the ethical approval granted by the institutional review board, public sharing is not permitted. Data access is therefore restricted to ensure compliance with confidentiality agreements and ethical standards.

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
