# Peer review of "Evidence-Based Practices to Prevent Ventilator-Associated Pneumonia in an Intensive Care Unit in Bangladesh"

_healthcare, 2025, doi:10.3390/healthcare13212782_

Round 1
Reviewer 1 Report
Comments and Suggestions for Authors
The manuscript “Effectiveness of Evidence-Based Practices to Prevent Ventilator-Associated Pneumonia in an Intensive Care Unit in a Tertiary Hospital in Bangladesh” presents a quasi-experimental study that evaluates the effectiveness of implementing a bundle of evidence-based practices (EBP) for the prevention of ventilator-associated pneumonia (VAP) in an intensive care unit at a tertiary hospital in Bangladesh. This is a relevant topic, especially in low- and middle-income countries, where there is a shortage of interational studies and high mortality rates in critically ill patients.
The title is clear and descriptive, but excessively long and redundant. It is recommended to simplify it, maintaining accuracy, but favoring clarity and impact.
The introduction is detailed and contextualizes the problem well, but in some passages there is an excess of narrative review that could be summarized. For example, the overlap of information on the incidence of VAP globally and in LMICs could be more concise
The justification for using the definition of clinically suspected VAP (lines 64–73) is pertinent, but needs to be more critical, better discussing the limitations of diagnostic validity.
The quasi-experimental non-randomized with historical control design is justified, but there is insufficient detail on potential temporal biases (seasonal changes, staff variation, infrastructure, etc.). The sample calculation is not presented; the text mentions an estimate of 450 patients (lines 128–129), but does not report whether statistical power was calculated. The instrument for assessing nurses' competencies (64-item checklist) lacks further details on the validation process, beyond the statement that validity was confirmed (lines 153–157).
The results are presented clearly, with well-structured tables. The reduction in the incidence of VAP was well presented, but the explanation for the increase in overall ICU mortality (from 50.9% to 62.6%) is insufficient. The result deserves more critical analysis, as it may indicate limitations of the intervention or selection bias. The tables are clear, but the flowchart (Figure 2) lacks detail on exclusions and losses, which may compromise transparency.
Although the discussion compares data with the international literature, the argument that the increase in mortality is due to “other critical care factors” is generic and needs further elaboration, for example, on antimicrobial resistance (supplementary data show a high prevalence of Acinetobacter and Klebsiella).
The initial hypothesis that nurse training would be a determining factor was partially confirmed, but the causal relationship is discussed in a superficial manner.
The manuscript mentions limitations (temporal difference between groups, manual data collection, lack of randomization). However, there is insufficient discussion of external validity (whether the findings can be extrapolated to other hospitals in Bangladesh or LMICs). The implications of using “clinically suspected VAP” outcomes rather than those confirmed by microbiological criteria are also not explored.
Most references are current and relevant (including recent systematic reviews and guidelines). However, some are outdated or redundant (e.g., American Thoracic Society; Infectious Diseases Society of America 2005 could be replaced or supplemented by more recent updates).
Author Response
Responses:
To Reviewer 1
Thank you very much for reviewing our manuscript and considering for further processing. Please find the following responses point-by-point and highlight in ‘Yellow’ all the changes in the main text.
Sincerely,
Nahida Akhtar
Hiroshima University
Comments 1. The title is clear and descriptive, but excessively long and redundant. It is recommended to simplify it, maintaining accuracy, but favoring clarity and impact.
Response 1: We revised the title and tried to simplify it to “Evidence-Based Practices to Prevent Ventilator-Associated Pneumonia in an Intensive Care Unit in Bangladesh” (page 1).
Comments 2. The introduction is detailed and contextualizes the problem well, but in some passages, there is an excess of narrative review that could be summarized. For example, the overlap of information on the incidence of VAP globally and in LMICs could be more concise.
Response 2: We revised throughout the ‘Introduction’ accordingly (Pages 1 - 2).
Comments 3. The justification for using the definition of clinically suspected VAP (lines 64–73) is pertinent, but needs to be more critical, better discussing the limitations of diagnostic validity.
Responses 3: Sorry for the inconvenience. We revised the definition of clinically suspected VAP to make it clearer and discussed more in the ‘Limitation’ part (Page 4, Line 130-137 and Page 09, Line 402-410).
Comments 4. The quasi-experimental non-randomized with historical control design is justified, but there is insufficient detail on potential temporal biases (seasonal changes, staff variation, infrastructure, etc.). The sample calculation is not presented; the text mentions an estimate of 450 patients (lines 128–129) but does not report whether statistical power was calculated. The instrument for assessing nurses' competencies (64-item checklist) lacks further details on the validation process, beyond the statement that validity was confirmed (lines 153–157).
Responses 4: We explained the reason why we chose this design (Page 2, Line 84-89).
Regarding potential temporal biases: we used same nurses and same ICU. But seasonal variation might be influenced that we mentioned in the ‘Limitation’ section (Page 10, Line 406-410).
As an exploratory design, we considered the estimated sample size based on the monthly patient admission within the study time period. This is explained in the text (Page 4, L120-123). As we not calculated the ‘Sample size’, it might be cause underpower of this study and we mentioned it in the ‘Limitation’ section (Page 10, line 401-402)
Content validity of nurses’ skills checklist (the tool) was added (Pages 4-5, line 154-173).
Comments 5. The results are presented clearly with well-structured tables. The reduction in the incidence of VAP was well presented, but the explanation for the increase in overall ICU mortality (from 50.9% to 62.6%) is insufficient. The result deserves more critical analysis, as it may indicate limitations of the intervention or selection bias. The tables are clear, but the flowchart (Figure 2) lacks detail on exclusions and losses, which may compromise transparency.
Responses 5: We explained the possible cause of death in pages 9-10, Lines 349-361 in the ‘Discussion’ part.
All the biases are mentioned in the ‘Limitation’ section (Page 10, Line 401-410).
Following your advice, we revised the flowchart based on the ‘TREND’ chart (Figure 2, Page 7).
Comments 6. Although the discussion compares data with the international literature, the argument that the increase in mortality is due to “other critical care factors” is generic and needs further elaboration, for example, on antimicrobial resistance (supplementary data show a high prevalence of Acinetobacter and Klebsiella).
Response 6: Based on the culture report, we described the death related to organisms on Page 9,10, Line 349-361.
Comments 7. The initial hypothesis that nurse training would be a determining factor was partially confirmed, but the causal relationship is discussed in a superficial manner.
Response 7: We apologize for the inconvenience. We could not determine the causal relationship between nurses' training and patients' outcomes, as all nurses were assigned for all (both intervention and control) patients (Page 2, Line 88-90). However, while there are limitations to the pre-post comparative study design for the nurses, the difference between the two periods is that “nurses received training on the VAP Bundle and implemented it for patients.” (Page 5, Line 166-168).
It is mentioned on
Comments 8. The manuscript mentions limitations (temporal difference between groups, manual data collection, lack of randomization). However, there is insufficient discussion of external validity (whether the findings can be extrapolated to other hospitals in Bangladesh or LMICs). The implications of using “clinically suspected VAP” outcomes rather than those confirmed by microbiological criteria are also not explored.
Response 8: We mention according to the ‘Limitation’ section on Page 10, Lines 401-410.
Comments 9. Most references are current and relevant (including recent systematic reviews and guidelines). However, some are outdated or redundant (e.g., American Thoracic Society; Infectious Diseases Society of America 2005 could be replaced or supplemented by more recent updates).
Response 9: We replaced the outdated reference as possible (ref no: 13).
Reviewer 2 Report
Comments and Suggestions for Authors
A well written article highlighting a common problem across the globe in the ICUs and the value of evidence based practices. WHO and CDC have highlighted the use of VAP bundles. The conclusions are contrary to these recommendations.
Few points are highlighted:
- The references are not as per Vancouver style
- VAP was defined only as per clinical suspicion without full microbiological and radiological confirmation. Yet bacteria were isolated in a few. How many cases in both the groups had bacterial isolation?
- It is not clear how were the control and intervention groups decided? Computer based division is mentioned but few points need clarification:
- Pl list the interventions in the control group
- How did the mortality and no of ventilator days increase in the intervention group?
- Why was prolonged mechanical ventilation
- Can the authors also comment on the cost difference between the two groups. increasing the use of closed section and the frequency of catheter use with along with the increase in manpower hours should also be worked out
Author Response
Responses:
To reviewer 2
Thank you very much for reviewing our manuscript and considering for further processing. Please find the following responses point-by-point and highlight in ‘Yellow’ all the changes in the main text.
Sincerely,
Nahida Akhter
Hiroshima University
Comments 1. A well written article highlighting a common problem across the globe in the ICUs and the value of evidence-based practices. WHO and CDC have highlighted the use of VAP bundles. The conclusions are contrary to these recommendations.
Response 1: We revised the ‘Conclusion’ and minimized all the contradictory statements (Page 11, Line 419-429).
Comments 2. The references are not as per Vancouver style
Response 2: We follow journal requirements as recommended by the ACS style guide.
Comments 3. VAP was defined only as per clinical suspicion without full microbiological and radiological confirmation. Yet bacteria were isolated in a few. How many cases in both groups had bacterial isolation?
Response 3: We isolated 9 different species of bacterial pathogens and discovered 321 total specimens: 165 in the control group and 156 in the intervention group. We described the distribution of the organism in the Supplementary file 1.
Comments 4. It is not clear how were the control and intervention groups decided? Computer based division is mentioned but few points need clarification: Pl list the interventions in the control group. How did the mortality and no of ventilator days increase in the intervention group? Why was prolonged mechanical ventilation
Responses 4: We apologize for the inconvenience. We revised ‘Figure 2; for more clarification.
For the intervention group, the prolonged mechanical ventilation was observed. We explained more in the discussion part (Page 9-10, Line 349-361).
Comments 5. Can the authors also comment on the cost difference between the two groups. increasing the use of closed section and the frequency of catheter use with along with the increase in manpower hours should also be worked out.
Responses 5: We did not calculate the cost difference between the two groups. As we used closed suction instead of open and increased the frequency of catheter use in the intervention group: we implemented these based on evidence-based practice. We used existing manpower (nurses), did not use any additional manpower for this study.
Reviewer 3 Report
Comments and Suggestions for Authors
Dear Authors,
Thank you for the opportunity to read and review your manuscript.
The study is well-structured and addresses a relevant topic. I think it may offer a valuable contribution to the evidence from a low-resource context (which is pretty unexplored).
The manuscript is well-structured. It is clearly written, the methodology is adequate, and the references are up to date and appropriate. The design is well justified given the setting limitations, and the statistical analysis is adequate.
The results are effectively contextualized in the discussion, highlighting the practical implications for nursing practice in resource-limited ICUs.
I have only two major concerns.
The first is related to the CONSORT checklist, which is specific for randomized studies. Please, evaluate if the TREND checklist is more suitable for your study design and attach the filled checklist as a supplementary.
The second is about the increase in ICU mortality during the intervention period (as reported in table 2 and in the results/discussion). I think that this finding requires a deeper interpretation, as it may be related to factors beyond the intervention itself.
Please, check also minor comments in the attached PDF file.
Sincerely

Author Response
Responses:
To reviewer 3
Thank you very much for reviewing our manuscript and considering for further processing. Please find the following responses point-by-point and highlight in ‘Yellow’ all the changes in the main text.
Sincerely,
Nahida Akhter
Hiroshima University
Comments 1. The first is related to the CONSORT checklist, which is specific for randomized studies. Please, evaluate if the TREND checklist is more suitable for your study design and attach the filled checklist as a supplementary.
Response 1: We replaced the CONSORT checklist to TREND checklist and attached as a ‘Supplementary File’.
Comments 2. The second is about the increase in ICU mortality during the intervention period (as reported in table 2 and in the results/discussion). I think that this finding requires a deeper interpretation, as it may be related to factors beyond the intervention itself.
Response 1: We added more explanation in the discussion part (Page 9-10, Line 349-361).
Comments 3. Please, check also minor comments in the attached PDF file.
(1) Replace the conceptual framework in the method section.
Response : We replaced it as Figure 1 accordingly.
(2) Spell out the abbreviation
Response: We did this accordingly. Other parts we change accordingly in the text.
Round 2
Reviewer 1 Report
Comments and Suggestions for Authors
Second evaluation of the manuscript “Evidence-Based Practices to Prevent Ventilator-Associated Pneumonia in an Intensive Care Unit in Bangladesh.” The authors satisfactorily addressed most of the previous recommendations, improving the clarity, structure, and methodological consistency of the study.
The title was simplified, making it more direct and informative. The introduction was edited, reducing redundancies and maintaining a solid contextualization of the problem in the setting of low- and middle-income countries. Important improvements were observed in the method: the justification for the quasi-experimental design was expanded, the potential temporal bias was recognized, and the limitation regarding the absence of formal sample size calculation was duly mentioned. Also noteworthy is the more complete description of the nursing skills assessment tool, now with evidence of validity and reliability, which strengthens the credibility of the findings.
The results were presented clearly and accompanied by well-organized tables. The revision of the flowchart according to the TREND guidelines contributes to greater transparency, although it requires occasional additions. The discussion is more closely linked to recent literature, and the authors recognized important limitations, such as the absence of randomization, possible underpowering, and seasonal variations. The inclusion of the supplement with microbiological data added interpretive value and reinforced the relevance of the topic in the context of antimicrobial resistance.
However, I believe that some points still require further review:
1) The explanation for the increase in overall mortality in the post-intervention period remains generic. It is recommended to include or discuss, in a more structured way, data on the baseline severity of patients (e.g., APACHE II, SOFA, or main diagnosis), or to explain its absence and acknowledge the possible confounding bias. If possible, it is suggested to perform an adjusted analysis with relevant clinical variables.
2) It is necessary to clarify whether there was a change in the criteria or frequency of microbiological sample collection between periods, since the supplement indicates higher culture positivity in the post-intervention period. This information is essential to correctly interpret the evolution of pathogens and mortality.
3) Since the same professionals participated in the training and data collection, it is recommended to discuss more clearly how this potential bias was minimized (e.g., use of independent supervision or cross-checking of data).
4) It is important to broaden the discussion on the applicability of the results in other hospitals in Bangladesh or in low- and middle-income countries, considering the structural and resource specificities of the studied setting.
5) It is recommended to moderate expressions that indicate a direct causal relationship between training and clinical outcomes, adopting language of association compatible with the quasi-experimental design.
Author Response
Reviewer 1:
Thank you very much for reviewing our manuscript and considering for further processing. Please find the following responses point-by-point and highlight all the changes in the main text.
Sincerely,
Nahida Akhter
Hiroshima University
1) The explanation for the increase in overall mortality in the post-intervention period remains generic. It is recommended to include or discuss, in a more structured way, data on the baseline severity of patients (e.g., APACHE II, SOFA, or main diagnosis), or to explain its absence and acknowledge the possible confounding bias. If possible, it is suggested to perform an adjusted analysis with relevant clinical variables.
Response: In this study, we have GCS data, and we showed the differences between the intervention and control groups in Table 2. We also tried to explore the high mortality in the intervention group and for that we used patients' main diagnosis and showed in Table 3. We added this statement in the ‘Limitation’ section (page 11, Line 440-442).
2) It is necessary to clarify whether there was a change in the criteria or frequency of microbiological sample collection between periods, since the supplement indicates higher culture positivity in the post-intervention period. This information is essential to correctly interpret the evolution of pathogens and mortality.
Response: The criteria and frequency of microbiological sample collection remained consistent between the pre- and post-intervention periods. The higher culture positivity in the post-intervention period may be due to by chance.
4) It is important to broaden the discussion on the applicability of the results in other hospitals in Bangladesh or in low- and middle-income countries, considering the structural and resource specificities of the studied setting.
Response: According to your suggestion, we added the statement on Page 11, Line 420-424.
5) It is recommended to moderate expressions that indicate a direct causal relationship between training and clinical outcomes, adopting language of association compatible with the quasi-experimental design.
Response: We apologize for the inconvenience. We could not establish the direct causal relationship between training and clinical outcomes as the nurses conducted roster duty (page 2, Line 88-89). All the study nurses were assigned to all the ICU patients (both intervention and control groups). We added this in the ‘Limitation’ section (page 11, Line 445-448).
“adopting language of association compatible with the quasi-experimental design”: we changed the statement on page 10, Line 364.

Reviewer 2 Report
Comments and Suggestions for Authors
The authors have taken cognisance of the comments of the Reviewers and have made substantial improvements in the article. It is a well structured article, conducted in a place with limited resources.
The authors have defined the study well (quasi-experimental, non-randomized, two-arm study design with a non-equivalent historical control group was conducted, comparing two sequential time periods)
However, few points still need to be addressed:
- “This may be explained by the significantly higher proportion of late-onset VAP associated with prolonged mechanical ventilation in the intervention group, along with limitations in overall patient management and challenges in reducing ventilation duration.”
The reason for higher proportion of VAP and longer duration of MV has not been explained. There was no randomisation: all patients were subjected to Interventional Bundles. So why should there be an increase in the duration of mechanical ventilation? Alternatively, was there increased mortality in the control group leading to a spurious lower duration of ventilation?
The high prevalence of multidrug-resistant (MDR) pathogens in the intervention patients, could be explained due to the prolonged period of ventilation.
- There is a discrepancy in line 35 and 140: “ VAP is defined as developing after 48 hours of intubation [1]” “…primary endpoint determined the incidence rate of VAP within 2 months”
Later the authors have written “Line 231…mortality rate of mechanically ventilated patients in ICU, early-onset (<96 hours) and late- onset (≥96 hours) of VAP, and the average length of stay of patients in ICU.
Author Response
Reviewer 2:
Thank you very much for reviewing our manuscript and considering for further processing. Please find the following responses point-by-point and highlight all the changes in the main text.
Sincerely,
Nahida Akhter
Hiroshima University
- “This may be explained by the significantly higher proportion of late-onset VAP associated with prolonged mechanical ventilation in the intervention group, along with limitations in overall patient management and challenges in reducing ventilation duration.”
The reason for higher proportion of VAP and longer duration of MV has not been explained. There was no randomization: all patients were subjected to Interventional Bundles. So why should there be an increase in the duration of mechanical ventilation? Alternatively, was there increased mortality in the control group leading to a spurious lower duration of ventilation?
The high prevalence of multidrug-resistant (MDR) pathogens in the intervention patients, could be explained due to the prolonged period of ventilation.
Response: We discussed ‘the most possible cause of death in the intervention group’ on page 10, Line 379-391.
- There is a discrepancy in line 35 and 140: “VAP is defined as developing after 48 hours of intubation [1]” “…primary endpoint determined the incidence rate of VAP within 2 months” Later the authors have written “Line 231…mortality rate of mechanically ventilated patients in ICU, early-onset (<96 hours) and late- onset (≥96 hours) of VAP, and the average length of stay of patients in ICU.
Response: We apologize for the inconvenience. We deleted ‘within 2 months’ (page 4, Line 140). VAP cases were diagnosed after 48 hours and early-onset (less than 96 hours) and late-onset (≥96 hours) we categorized by the CDC [https://www.cdc.gov/nhsn/pdfs/pscmanual/6pscvapcurrent.pdf].
